# A Conserved Di-Lysine Motif in the E2 Transactivation Domain Regulates MmuPV1 Replication and Disease Progression

**DOI:** 10.3390/pathogens14010084

**Published:** 2025-01-16

**Authors:** Jessica Gonzalez, Marsha DeSmet, Elliot J. Androphy

**Affiliations:** 1Department of Microbiology and Immunology, Indiana University School of Medicine, Indianapolis, IN 46202, USAmdesmet@iu.edu (M.D.); 2Department of Dermatology, Indiana University School of Medicine, Indianapolis, IN 46202, USA

**Keywords:** papillomavirus, HPV, MmuPV1, E2, post-translational modification

## Abstract

The papillomavirus E2 protein regulates the transcription, replication, and segregation of viral episomes within the host cell. A multitude of post-translational modifications have been identified which control E2 functions. A highly conserved di-lysine motif within the transactivation domain (TAD) has been shown to regulate the normal functions of the E2 proteins of BPV-1, SfPV1, HPV-16, and HPV-31. This motif is similarly conserved in the E2 of the murine papillomavirus, MmuPV1. Using site-directed mutagenesis, we show that the first lysine (K) residue within the motif, K112, is absolutely required for E2-mediated transcription and transient replication in vitro. Furthermore, mutation of the second lysine residue, K113, to the potential acetyl-lysine mimic glutamine (Q) abrogated E2 transcription and decreased transient replication in vitro, while the acetylation defective arginine (R) mutant remained functional. Both K113 mutants were able to induce wart formation in vivo, though disease progression appeared to be delayed in the K113Q group. These findings suggest that acetylation of K113 may act as a mechanism for repressing MmuPV1 E2 activity.

## 1. Introduction

Papillomaviruses (PVs) are non-enveloped, double-stranded DNA viruses that infect the stratified epithelia. There are over 200 different types of human papillomavirus which can be broadly categorized as low-risk or high-risk. Long-term infection with a high-risk HPV is associated with cancers of the oropharynx and anogenital tract, and is the causative agent of nearly all cervical cancers [1]. PVs have a wide host range yet are species-specific [2], a fact that has historically made study of the complete viral life cycle difficult.

The PV life cycle is tightly linked to the differentiation of the host cell and can be divided into three stages: establishment, maintenance, and vegetative amplification. Infection with a PV begins when a break in the host epithelium allows incoming virions access to the basement membrane [3]. Upon binding the basement membrane, virions enter the basal cells and are trafficked intracellularly to the host nucleus where the virus replicates to a low copy number to establish infection. After a pool of viral genomes is established, the virus enters the maintenance phase of infection, during which the viral genome replicates in synchrony with the host cell to maintain multicopy episomes during cell division. In the suprabasal layers of the epithelium, the virus transitions into the vegetative amplification stage of infection, in which the L1 and L2 capsid proteins are expressed, the viral genome replicates to a high copy number, and progeny virions assemble [4]. The viral life cycle is tightly controlled, and failure to navigate the stages can lead to loss of viral episomes during cell division or integration of the viral genome into the host genome. Integration is often a first step in PV-induced carcinogenesis [5], so elucidation of the molecular mechanisms which control the transition between viral life cycle stages may offer insight into PV-associated carcinogenesis.

The viral E2 protein is a multifunctional regulatory protein that controls viral transcription, replication, and segregation of viral episomes in dividing cells. The structure of E2 consists of an N-terminal transactivation domain (TAD) and a C-terminal DNA binding and dimerization domain (DBD) connected by a flexible hinge region [6]. The sequence of the TAD is highly conserved, and several amino acids within this domain are critical for E2 function [7,8]; among these critical amino acids are two consecutive lysine residues, typically found at positions 111 and 112. This di-lysine motif is present in many papillomavirus types, and mutational analyses have indicated that acetylation of the first lysine residue in this motif is a major mechanism for regulating E2 activity [9,10,11,12,13].

Several studies have characterized acetylation-defective arginine substitutions in this di-lysine motif to elucidate how acetylation regulates E2 activity. In BPV-1, acetylation of the first lysine by p300 enhances E2-dependent transcription and promotes E2 nuclear localization; mutation of this lysine to arginine lead to severe transcriptional defects that could not be enhanced by p300 expression [9]. In both BPV-1 and HPV-31 E2, mutation of the first lysine to arginine abrogates transient DNA replication; in BPV-1, mutation of the second lysine also reduces transient replication, though to a lesser extent than mutation of the first lysine [10]. The replication defect identified in HPV-31 E2 is attributed to a failure of the acetylation-deficient mutant to recruit topoisomerase 1 to the viral origin [11] preventing origin unwinding [10], indicating that acetylation by p300 regulates E2 replication functions. In HPV-16, acetylation of the first lysine by p300 regulates E2 stability during mitosis by preventing ubiquitination on the second lysine in the motif [12]. In the cottontail rabbit papillomavirus SfPV1, mutation of the first lysine to the acetyl-lysine mimic glutamine reduced transcription activation due to reduced binding to the C-terminal motif (CTM) of Brd4 [13].

In this study, we utilized a mutational approach to investigate the E2 di-lysine motif in the murine papillomavirus, MmuPV1. MmuPV1 is the first PV discovered to infect the laboratory mouse *Mus musculus*, offering an attractive model for investigating E2 functions both in vitro and in vivo in the natural host species [14]. MmuPV1 can productively infect both cutaneous and mucosal sites [15] and can cause cancer in either type of epithelium [16]. Site-directed mutagenesis was performed to generate either acetyl-lysine mimetic glutamine (Q) or acetylation-defective arginine (R) substitutions within each residue of the di-lysine motif. Both mutations of the first lysine, K112, rendered E2 completely non-functional in transcription and transient replication assays, but did not alter subcellular localization. The acetylation defective mutant of the second lysine, K113R, was able to transiently replicate and activate transcription, though not as strongly as WT E2. In contrast, K113Q displayed significant defects in transcription and transient replication compared to WT E2, which is likely attributable to interference with normal E2 protein–protein interactions. Despite these defects displayed by K113Q in vitro, each of the K113 mutants induced wart formation in immunocompromised mice, though K113Q appeared to cause a delay in disease progression, which may be reflective of the poor replication in transient assays. The work presented here suggests that acetylation of K113 may act as a mechanism to repress E2 activity without completely abolishing function of the protein.

## 2. Materials and Methods

Cell culture and transfections. All cells were maintained in 5% CO_2_ at 37 °C. HEK293TT, C33-A, and CV-1 cells were cultured in Dulbecco’s Minimal Essential Medium (Life Technologies, Carlsbad, CA, USA) supplemented with 10% fetal bovine serum (Peak Serum, CO, USA) and 1% penicillin/streptomycin (Life Technologies). C33-A and CV-1 transfections were performed using Lipofectamine 2000 (Invitrogen, Carlsbad, CA, USA) according to manufacturer instructions. HEK293TT transfections were performed using 2 μg/mL polyethyleneimine (PEI) at a ratio of 2:1 PEI:μg DNA [17]. All transfection reactions were performed in Opti-MEM (Life Technologies). Empty vector DNA (either pcDNA3 or pCI-V5) was included wherever necessary to keep DNA concentrations consistent across transfections within the same experiment.

Plasmids and antibodies. The pMusPV1 plasmid was provided by John Schiller [18]. The pUC19-E2, pCI-V5-E2, pCI-V5-E2 TTL, pCI-myc-E1, pCI-myc-E1 TTL, and mFLori plasmids were described previously [19]. The pGL2-4xE2BS-Luc plasmid was described previously [20]. Site-directed mutagenesis was performed on pUC19-E2 with the Q5 Site-Directed Mutagenesis Kit (NEB, Ipswich, MA, USA #M04925) using the mutagenic primers K112Q Fwd: GTATGGGACTTTTCAAAAAAGTGGCG, K112R Fwd: GTATGGGACTTTTAGAAAAAGTGGCG, K112 Rev: GGCTCAGTGCTGTCATAC, K113Q Fwd: GGACTTTTAAACAAAGTGGCGAGG, K113R Fwd: GGACTTTTAAACGAAGTGGCGAGG, and K113 Rev: CATACGGCTCAGTGCTGT. Mutant E2 ORFs were PCR amplified from pUC19-E2 using the primers V5-E2 Fwd: ATGCGAATTCATGAACAGCCTGGAAACACGTTT and V5-E2 Rev: ATGCGTCGACTCAGAGTCCGTCTAAGAAGC. PCR products and the pCI-V5 vector were digested with EcoRI and SalI prior to ligation to generate the mutant pCI-V5-E2 expression vectors.

The mutant viral genomes were generated by amplifying the E2 mutation from pUC19-E2 using primers E1 Fwd: AAGCATAGGAAGCCAGTGCG and E2 Rev: TGATGCCGGCTGTTGTTTGG. PCR products and the pMusPV1 plasmid were digested with Eco81I and MluI (Invitrogen) prior to ligation. Sequencing was performed at all steps of cloning to ensure the E2 mutations were retained and no off-target mutations were generated.

Mouse anti-V5 antibody (Invitrogen #R960-25), rabbit anti-V5 antibody (Cell Signaling #13202), rabbit anti-Myc-tag (Cell Signaling, Danvers, MA, USA #2278), mouse anti-α tubulin (Sigma-Aldrich #T6199), mouse anti-GAPDH (Santa Cruz, Dallas, TX, USA #sc-47724), AlexaFluor 488 conjugated goat anti-mouse (Invitrogen #A-11029), and AlexaFluor 594 conjugated goat anti-rabbit (Invitrogen #A-11012) were used as indicated. Antibody dilutions are specified in figure legends.

Transcription assays. C33-A cells in a 6-well plate were transfected with 200 ng of an E2-responsive Firefly luciferase reporter containing four high-affinity E2 binding sites, pGL2-4xE2BS-Luc, and 100 ng of expression vectors for WT or mutant pCI-V5-E2. Approximately 48 h post-transfection cells were lysed and luciferase assays were performed as outlined by manufacturer instructions (Promega, Madison, WI, USA). Luminescence was measured on a PHERAstar plate reader and values were calculated as fold change over cells expressing the reporter in the absence of E2, to account for background luminescence. Significance was determined by paired *t*-tests between the WT and mutant samples.

For over-expression assays, transfections were performed in a 12-well plate using 100 ng of pGL2-4xE2BS-Luc and increasing amounts of the WT and mutant pCI-V5-E2 expression vectors (1×: 50 ng, 5×: 250 ng, 10×: 500 ng). The empty pCI-V5 vector was used in the reporter only, 1×, and 5× samples to maintain constant DNA concentrations across each transfection. Firefly luciferase assays were performed as above approximately 48 h post-transfection. Values were calculated as above. Significance was determined as indicated in figure legends.

Transient DNA replication. Transient replication assays were performed on C33-A cells in 96-well plates as described previously [10,21]. Each well was transfected with mFlori (5.6 ng) and pRLuc (1.4 ng) and either mutant or WT pCI-V5-E2 (14 ng). Each E2 variant was co-transfected with either full-length pCI-myc-E1 or an early termination E1 mutant (28 ng). Approximately 72 h post-transfection cells were lysed and assessed by Dual-Glo Luciferase Assay (Promega). Firefly and Renilla luciferase luminescence were measured on a PHERAstar plate reader. Values were calculated as the ratio of Firefly over Renilla luminescence and normalized to cells containing only the reporters in the absence of E2. Assays were performed a minimum of three separate times with eight technical replicates each. Significance was determined by paired *t*-test between samples with and without full-length E1.

Immunofluorescence. HEK293TT or CV-1 cells were plated on sterile glass coverslips in a 6-well dish and transfected with 3 μg of WT or mutant pCI-V5-E2 expression vectors. Untransfected cells or cells transfected with mCherry were used as negative controls. Approximately 48 h post-transfection, media were removed, cells were washed with 1× PBS and then fixed with 4% paraformaldehyde at room temperature. After fixation, cells were permeabilized with 0.1% Triton-X100 in PBS for 10 min at room temperature. Cells were then incubated in blocking solution (PBS + 0.1% Triton-X100 + 10% normal goat serum) for approximately 1 h at room temperature. Primary anti-V5 antibody was added to the blocking buffer at the indicated dilution and cells were incubated rocking overnight at 4 °C. Primary antibody was removed, cells were washed thrice in 1× PBS and then incubated with the indicated AlexaFluor-conjugated secondary antibody diluted in blocking buffer for approximately 2 h at room temperature, rocking and protected from light. Secondary antibody was removed, cells were washed three times in 1xPBS, and coverslips were mounted on microscope slides with ProLong Gold with DAPI (Invitrogen). Slides were allowed to cure in the dark for at least 24 h prior to imaging with a Nikon Eclipse 80 i microscope at 40× magnification.

Immunoblotting. Whole cell lysates were separated by SDS-PAGE on ExpressPlus PAGE, 4–12% gradient gels (Bio-Rad, Hercules, CA, USA) and transferred to 0.45μM PVDF membranes (Millipore) using the semi-dry transfer method. Membranes were blocked with 5% nonfat milk in PBST for at least 45 min at room temperature. Blocking solution was removed, membranes were washed at least three times in PBST, and then membranes were incubated with primary antibodies diluted in PBST overnight at 4 °C. E2 was blotted with anti-V5 antibodies as indicated in figure legends; loading controls were blotted with either mouse anti-GAPDH or mouse anti-α-tubulin as indicated. Myc-E1 was blotted with rabbit anti-Myc tag (71D10).

Co-immunoprecipitation. HEK293TT cells were transfected with expression vectors for WT, K113Q, or K113R V5-E2 and pCI-myc-E1. To compensate for poor expression of the K113 mutants, these transfections were performed with 1.5× the E2 vector amount used in WT transfections. The empty vector pcDNA3 was used as carrier DNA to maintain equivalent DNA concentrations across transfections. Approximately 30 h post-transfection, cells were washed with ice-cold PBS and incubated on ice in IP lysis buffer (50 mM Tris, pH 7.2; 150 mM NaCl; 0.1% NP-40; 1 mM DTT; protease inhibitor cocktail) for 30 min. Cells were collected by scraping and centrifuged at 16,000× *g* for 30 min at 4 °C. Supernatant was moved to a new tube and the insoluble fraction was discarded. Approximately 10% of the supernatant was reserved for input; the remainder of the supernatant was incubated overnight with anti-c-myc magnetic beads (MedChemExpress, Monmouth Junction, NJ, USA HY-K0206) on a rotator at 4 °C. Beads were pre-blocked in 2% BSA and washed 3× in PBS prior to use. After overnight incubation, beads were washed 5× in IP wash buffer (50 mM Tris, pH 7.5; 170 mM NaCl; 0.1% NP-40; protease inhibitor cocktail) and 1× in IP lysis buffer. Beads were incubated in 1× Laemmli buffer at 65 °C for 5 min to elute the bound protein. Protein complexes were separated by SDS-PAGE and immunoblotted as described above.

DNA isolation and PCR. DNA was isolated from tissue samples using the DNeasy Blood & Tissue Kit (Qiagen, Hilden, Germany). PCR was performed on isolated DNA using the following primers: m1800 Fwd: TGTTGGGTCATCATCGTTGT, m3300 Rev: GGAAGTTTGCAATAACCTTCCAGTCCC, m3000 Fwd: GTAGTATGGTGCAGTGGGCA, and mE2-L2 Rev: ATCGGGGGTCACCTCAAG. PCR products were purified using the GFX PCR DNA and Gel Band Purification Kit (Cytiva, Marlborough, MA, USA). Purified PCR product was sequenced to confirm presence of the K113 mutations in the tissue samples and to screen for off-target mutations within the E1 and E2 ORFs.

In vivo wart formation. All mice were maintained in the Indiana University School of Medicine Laboratory Animal Resource Center (LARC) in compliance with international standards and approval under IU School of Medicine Animal Care and Use Committee (IACUC) protocol #23035. All experiments were performed in homozygous Hsd:Athymic Nude-Foxn1^nu^ mice obtained from Inotiv (West Lafayette, IN, USA). Circularized mutant viral genomes were prepared as described previously [22]. Beginning one week prior to injection, mice were started on water supplemented with estradiol (8 μg/mL) and were maintained on estradiol for approximately 3 months to increase MmuPV1 copy number [23]. Mice were anesthetized with isoflurane and wounded on the tail with sterile sandpaper approximately 72 h prior to injection, as described previously [15,24]. Each mouse was injected with approximately 20 μg of circular mutant viral genome intradermally in the pre-wounded site on the tail using an insulin syringe. Mice were monitored weekly for development of cutaneous disease for 4.5 months. At the conclusion of the study, mice were humanely euthanized, pictures were taken, and tissue was collected for further analysis. Tissue sections containing lesions were formalin-fixed and paraffin-embedded for hematoxylin and eosin (H&E) staining at the IU School of Medicine Histology and Histomorphometry Core facility. Stained tissue sections were imaged with an Echo Revolve microscope and analyzed for markers of PV-induced disease, including hyperproliferation and koilocytosis.

Statistical analysis. All data were generated from a minimum of three independent assays. Statistical analysis was performed as specified using Microsoft Excel.

## 3. Results

### 3.1. K112 Is Essential for MmuPV1 E2 Activity

The TAD di-lysine motif is conserved across many papillomavirus types, including high-risk human papillomaviruses and several animal papillomaviruses, including the murine papillomavirus MmuPV1 (Figure 1A). Of note, while many PVs retain this motif at positions 111 and 112, in MmuPV1, the lysine residues are located at 112 and 113, though this does not affect the residues’ relative positions within the molecular structure (Figure 1B). To investigate the functional role of the conserved di-lysine motif in the context of MmuPV1, site-directed mutagenesis was performed to generate acetyl-lysine mimetic (K112Q and K113Q) and acetylation defective mutants (K112R and K113R). Mutations were confirmed by sequencing. In prior studies, mutation of the first lysine in the motif resulted in differential effects on E2 function, suggestive of regulation by lysine acetylation. We thus hypothesized that the homologous lysine (K112) in MmuPV1 E2 would similarly display differential phenotypes when mutated to arginine or glutamine.

The K112 mutants were assessed for transcriptional activation in C33-A cells using the pGL2-E2BS-Luc reporter, which contains four closely spaced E2 binding sites upstream of a promoter and gene for Firefly luciferase [20]. Cells were transfected with the luciferase reporter alone or in conjunction with V5-tagged expression vectors for WT or K112 mutant E2. Luminescence was measured as a surrogate for the transcriptional activity of each sample, quantified as the fold change relative to the reporter alone; the early termination mutant V5-E2 TTL (for translation termination linker) was used as a negative control for E2-mediated transcription. Neither K112Q nor K112R were able to activate transcription (Figure 2A) and this activity was not rescued by over-expression of the mutant vectors (Figure 2B). In contrast, over-expression of the WT V5-E2 vector led to a dose-dependent increase in transcriptional activation. Whole cell lysates from the over-expression assay were separated by SDS-PAGE and immunoblotted for V5-E2 to confirm that the K112 mutants were being expressed. Both K112Q and K112R demonstrated reduced protein levels compared to WT E2 at equivalent vector concentrations, though they did display dose-dependent increases in protein expression, so we concluded that poor protein expression could not explain the transcriptional defect of the K112 mutants (Figure 2C).

The K112Q and K112R mutants were next assessed for transient replication using the dual luciferase transient replication assay developed by the Archambault group [21]. In this assay, the PV origin of replication is inserted into a Firefly luciferase reporter. Co-transfection with expression vectors for E1 and E2 leads to replication of the Firefly luciferase reporter, resulting in an increased luminescence signal. Cells are also transfected with a Renilla Luciferase reporter which lacks the PV ori and E2 binding sites, thereby acting as a control for transfection efficiency. Replication is reported as the ratio of Firefly luciferase luminescence over the Renilla luciferase luminescence, and these ratios are then normalized to samples containing only the reporters to account for background luminescence. Comparisons are made between samples containing both the E2 and WT E1 expression vectors and samples containing the E2 expression vector and an early termination E1 vector to account for luminescence resulting from E2-mediated transcriptional activity on the Firefly reporter. We found that neither K112 mutant could support transient replication (Figure 2D).

E2 is normally located within the nucleus of the host cell. One of the nuclear localization signals (NLSs) in BPV-1 E2 is located within the TAD and overlaps with the di-lysine motif [26]. Because both K112 mutants were deficient for transcription and transient replication, it was possible that mutation of this lysine interfered with E2 localization. Immunofluorescence was performed on HEK293TT cells transfected with K112 mutant V5-E2 expression vectors. Untransfected cells and cells transfected with WT V5-E2 were used as the negative and positive controls, respectively. Both mutants successfully localized to the nucleus (Figure 2E), so mis-localization of the mutant proteins does not explain the functional defects demonstrated by the K112 mutants.

Taken together, our results suggest that mutation of K112 severely impairs E2 function. This impairment cannot be fully explained by a failure of protein expression or localization. Overall, we conclude that the first lysine in the TAD di-lysine motif is critical for MmuPV1 E2 function.

### 3.2. K113 Mutations

In previous studies of the di-lysine motif in E2, mutation of the first lysine yielded strongly differential results suggesting that acetylation of this residue regulates E2 activity. In the cottontail rabbit papillomavirus SfPV1, mutation of the first lysine yielded results similar to those we have observed for MmuPV1, in which mutation to either arginine or glutamine completely abrogated E2 transcriptional activity, though this was attributed to mis-localization to the cytoplasm [13]. In BPV-1 E2, while mutation of the first lysine had greater effects on E2 function, mutation of the second lysine caused moderate defects in transcription and replication [9,10], suggesting that this residue does hold some functional significance. Because the experiments on our K112 mutants suggest that the first lysine is critical for MmuPV1 E2 function, we next investigated how mutation of the second lysine, K113, regulates E2 function.

We repeated the over-expression transcription assay as described above, this time using mutants K113Q and K113R. We found that K113Q was severely deficient for activating transcription even when over-expressed ten-fold (Figure 3A), while K113R displayed dose-dependent transcriptional activation. Though K113R was able to activate transcription, it was significantly impaired when compared to WT E2 activity at the same expression level (Figure 3B). SDS-PAGE was performed on cell lysates over-expressing the K113 mutants (Figure 3C), which showed that both were expressed in a dose-dependent manner, though each mutant had slightly diminished protein levels compared to WT V5-E2. Under-expression of the mutant protein can account for the moderate transcription defect displayed by K113R but cannot explain the near-complete loss of transcriptional activation in K113Q.

### 3.3. K113Q Is Moderately Deficient for Transient Replication

In BPV-1, mutation of the second lysine to arginine caused a significant deficit in transient replication, while in HPV-31, this mutation did not affect transient replication [10]. To investigate the hypothesis that mutations of the second lysine would lead to replication defects, we repeated the transient replication assays using the K113Q and K113R mutants. Both mutants were able to support transient replication (Figure 4A). K113R was able to transiently replicate to levels comparable to WT E2, though K113Q displayed a statistically significant defect compared to WT (Figure 4B).

Because K113Q demonstrated defects in both transcription and transient replication, we hypothesized that this mutant may display poor nuclear localization. Immunofluorescence was performed on CV-1 cells transfected with WT or mutant V5-E2 vectors; the K112 mutant vectors were included to confirm their nuclear localization in a second cell type. Cells transfected with mCherry were used as an E2-negative control to demonstrate diffuse localization. All four mutants were localized to the nucleus (Figure 5); therefore, we concluded that mutation of the di-lysine motif does not alter E2 localization.

### 3.4. Mutation of K113 Affects E1 Binding

In HPV-31, mutation of the first lysine did not significantly affect E1 binding despite the observed replication defect displayed by K111R [10]. In HPV-16 E2, however, mutation of the second lysine to alanine (A) displayed a moderate defect in E1 binding [27]. MmuPV1 E2 K113 mutants were assessed for E1 binding by co-immunoprecipitation; because of the defect in transient replication (Figure 4B), we hypothesized that K113Q would demonstrate decreased binding to E1. Lysates from HEK293TT cells transfected with expression vectors for myc-E1 and WT or K113 mutant V5-E2 were immunoprecipitated with myc-conjugated magnetic beads. To account for the decreased expression of the K113 mutants (Figure 3C), cells were transfected with 1.5× the vector concentration used in WT E2 transfections. K113R was immunoprecipitated with E1, indicating that the mutant maintained E1 binding capability; K113Q displayed a moderate E1 binding defect (Figure 6), though complete abolishment of the interaction could not be confirmed.

### 3.5. Both K113 Mutants Induce Wart Formation In Vivo

Because both K113Q and K113R demonstrated at least moderate levels of transient replication in vitro, we proceeded to investigate if these mutants would be able to cause cutaneous disease in vivo. Approximately 20 μg of circularized viral genomes were injected intradermally in the tails of athymic hairless mice, as described previously [15,24,28]. Mice were monitored weekly for development of cutaneous lesions. A time-matched experimental group utilizing the WT MmuPV1 genome was not included to minimize risk of disease spread within the animal facility.

Approximately 2.5 months post-injection four of four mice in the K113R group had developed cutaneous lesions near the injection site on the tail; one mouse from this group was sacrificed at this time point to confirm the presence of the K113 mutation by sequencing. In contrast, only one of three mice in the K113Q group had developed a tail lesion at this time point. By 4 months post-injection, all mice in the K113R group had developed lesions on the tail and were developing small lesions on the muzzle, indicating secondary infection; at this same time point all mice in the K113Q group had developed tail lesions near the injection site but none had developed secondary infections on the muzzle. The time course of the in vivo experiments is summarized in Figure 7A.

Mice were sacrificed approximately 4.5 months post-injection. At this final time point, all mice in the K113R group had lesions on the tail and muzzle. All mice in the K113Q had developed lesions on the tail near the injection site, and at least one of three mice had developed a lesion on the muzzle (Figure 7B). Tissue samples were collected for further analysis. DNA was isolated from lesions collected from mice in each group and sequenced to confirm the presence of the K113 mutations (Figure 7C). Tissue sections containing warts were formalin-fixed and paraffin-embedded for hematoxylin and eosin staining (Figure 8). Tissue sections lacking visible lesions were used as normal tissue controls. Lesions from both experimental groups demonstrated profound histologic changes that are characteristic of PV infection, including hyperkeratosis, parakeratosis, and development of koilocytes. Together, these findings confirm that K113Q and K113R MmuPV1 genomes retain the ability to cause dysplastic cutaneous disease in vivo, suggesting that mutation of K113 does not prevent the productive stage of the viral life cycle.

## 4. Discussion

The di-lysine motif in the E2 TAD is very highly conserved across papillomavirus types and has been demonstrated to be critical for E2 function. Mutation to arginine (acetylation defective) or glutamine (acetyl-lysine mimetic) implied that acetylation of this motif differentially regulates E2 function in BPV-1 [9], HPV-16 [12], HPV-31 [10,11], and SfPV1 [13]. In most studies characterizing this motif, it is the first lysine that demonstrates acetylation-driven regulation of E2 activity. Here we have demonstrated that in MmuPV1 E2 mutation of the first lysine, K112, to either glutamine or arginine completely abrogated E2 transcription (Figure 2A,B) and transient DNA replication functions (Figure 2D). Both mutants were expressed in a dose-dependent manner (Figure 2C) and localized to the nucleus (Figure 2E), and we therefore conclude that the first lysine in the motif is necessary for E2 function. Both mutants demonstrated slight under-expression compared to WT E2 (Figure 2C), which may indicate diminished stability of the mutant proteins. E2 can be stabilized when it binds the viral E1 [29] or host Brd4 proteins [30,31,32], so we speculate that mutation of K112 alters some binding interface that influences E2 stability, thus explaining the slight expression defect as well as the complete functional defects demonstrated here.

We next assessed the second lysine in this motif, K113. The potential acetyl-lysine mimic K113Q was unable to activate transcription (Figure 3A) and displayed a moderate defect in transient replication (Figure 4B). In contrast, the acetylation deficient K113R could support transcription and transient replication (Figure 4B), though transcriptional activation was significantly decreased compared to WT E2 (Figure 3B). This indicates acetylation of K113 is not required for E2 activity. Like the K112 mutants, both K113 mutants had slightly decreased protein levels when compared to WT E2 but were expressed in a dose-dependent manner (Figure 3C).

All mutants of the MmuPV1 di-lysine motif were localized to the nucleus (Figure 5). In BPV-1 [9], SfPV1 [13], and HPV-16 [33] mutation of this motif did result in slightly altered localization, but this was not seen with HPV-31, though mutants of the second lysine were not assessed for localization in this study [10]. In the case of SfPV1, co-expression with E1 induced nuclear localization of the E2 mutants and restored transient replication activity of the arginine and alanine substitution mutants but not the glutamine mutant, which was determined to be due to a defect in binding the Brd4 C-terminal motif (CTM). Though binding to the Brd4 CTM was not assessed in the present study, the decreased protein levels compared to WT (Figure 3C) suggest that this stabilizing interaction may be partially inhibited by mutation of K113. The canonical binding interface for E2:CTM interaction is not known to include the TAD di-lysine motif [34] but mutation of these sites could induce a structural change that alters the interaction surface.

In HPV-31 mutation of the first lysine altered E2 replication functions; the acetyl-lysine mimic upregulated transient replication by increasing topoisomerase 1 (Topo1) recruitment to the origin of replication to facilitate origin unwinding. In contrast, the acetylation-deficient arginine was deficient for transient replication due to an inability to recruit Topo1 to the origin, resulting in a failure of origin unwinding [10,11]. Notably, each of the HPV-31 E2 mutants were still able to bind E1 and recruit it to the origin of replication. In contrast, none of the lysine mutants in MmuPV1 E2 displayed an increase in replication (Figure 2D and Figure 4A,B). Additionally, we found that K113Q (Figure 6) is defective for E1 binding. The initiation of viral replication begins with E2 binding E1 and recruiting it to the origin, after which E2 is displaced, E1 forms a double hexamer, and unwinding of the origin allows replication machinery access to the viral DNA. Because E2 binding E1 precedes all other steps for initiation of replication, we believe the transient replication defect of K113Q is explained by poor E1 binding, which would inhibit all downstream steps in the initiation of replication from proceeding efficiently.

It has been demonstrated in HPV-16 E2 that acetylation on the first lysine blocks ubiquitination on the second lysine, thereby stabilizing the protein during mitosis [12]. Lysine ubiquitination is a common step in signaling pathways [35], and cross-talk between lysine acetylation and ubiquitination is well-known to regulate protein stability [36], so it is likely that MmuPV1 E2 is indeed regulated in this way. In interphase cells, we did not observe increased stability of E2 protein, which might have been expected as the K113 mutants are not susceptible to ubiquitination. Indeed, all four mutants of the TAD di-lysine motif displayed decreased protein levels relative to WT E2. It is possible that a potential stabilization effect would become apparent if cells were synchronized in mitosis, but this has not yet been performed. Because the E2 K113 mutant formed infectious virus and formed warts, we can conclude that acetylation of this amino acid is not necessary for the full viral replicative program in vivo.

Despite the in vitro deficits exhibited by K113Q, both mutants were able to induce cutaneous disease in athymic hairless mice (Figure 7B). The development of secondary infections on the muzzle establishes that both mutants can replicate and package the viral episome to produce infectious virus. Mice challenged with K113R genomes developed cutaneous lesions at the injection site earlier than mice challenged with K113Q genomes, and secondary infection of the muzzle was similarly delayed in K113Q mice (Figure 7A). This trend may suggest that moderate defects in transient replication in vitro correlate with delayed wart formation.

Previous work on the E2 di-lysine motif in BPV-1, HPV-16, and HPV-31 E2 found that mutation of the first lysine to the acetylation-defective arginine leads to loss of E2 functions, including transcription, DNA replication, and association with mitotic chromatin, while the acetyl-lysine mimic tends to remain functional and even increased activity in the case of HPV-31. This trend is notably reversed in the cottontail rabbit papillomavirus, SfPV1, in which the acetyl-lysine mimic lost transcriptional activity. The studies presented here demonstrate a similar trend, in which the acetyl-lysine mimic K113Q is defective for E2-dependent transcription and transient replication in vitro. Co-immunoprecipitation data suggest that a defect in E1 binding can explain the replication defects observed in vitro and in vivo; we hypothesize that disruption of additional protein interactions could explain the transcription defect and decreased protein stability of K113Q. Overall, our results suggest that acetylation on K113 may have a moderately inhibitory effect on normal MmuPV1 E2 functions, potentially as a means to navigate the stages of the viral life cycle while avoiding immune detection by the host.

## Figures and Tables

**Figure 1 pathogens-14-00084-f001:**
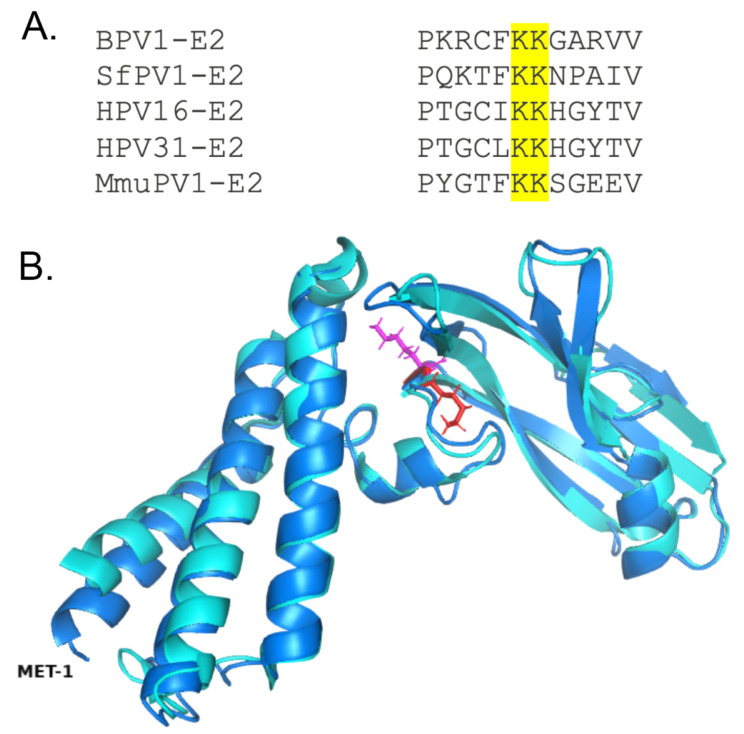
The TAD di-lysine motif is highly conserved. (**A**) Multiple sequence alignment of BPV-1, SfPV1, HPV-16, HPV-31, and MmuPV1 E2 protein sequences. The di-lysine motif (highlighted in yellow) is conserved across papillomavirus types. (**B**) Structure alignment between the solved BPV-1 TAD (cyan, PDB 2JEU) and the predicted MmuPV1 TAD (dark blue). The first lysine of the motif is depicted in red, and the second lysine is depicted in magenta. Structure prediction was performed with I-TASSER-MTD [25]; alignment and visualization were performed in PyMOL.

**Figure 2 pathogens-14-00084-f002:**
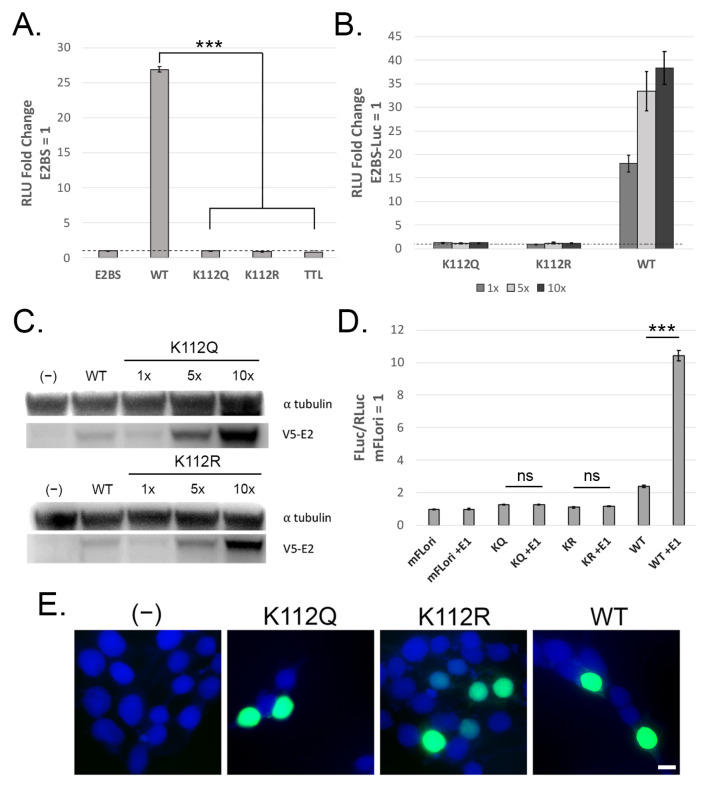
The first lysine in the di-lysine motif is critical for MmuPV1 E2 function. (**A**) Mutants of the first lysine, K112, fail to activate transcription from the pGL2-4xE2BS-Luc reporter. (**B**) Transcriptional activity is not restored by over-expression of the K112 mutants. WT E2 activates transcription in a dose-dependent manner. pCI-V5-E2 plasmid amounts were as follows: 50 ng (1×), 250 ng (5×), 500 ng (10×). (**C**) Immunoblot of cell lysates from panel C shows that K112 mutants are expressed in a dose-dependent manner. Mouse anti-V5 (1:3000), mouse anti-α-tubulin (1:10,000). (**D**) Neither K112Q nor K112R can support transient replication in the presence of E1. WT E2 demonstrates robust transient replication when co-expressed with WT E1. (**E**) Immunofluorescence was performed in HEK293TT cells transfected with expression vectors for WT or K112 mutant V5-E2. Both mutants are present within the nucleus, indicating that mutation of K112 does not interfere with nuclear localization of MmuPV1 E2. Mouse anti-V5 (1:1500), AlexaFluor 488 anti-mouse (1:3000). Nucleus is stained with DAPI. *n* = 3, *** *p* < 0.001, paired *t*-test. Scale bar represents 10 μM.

**Figure 3 pathogens-14-00084-f003:**
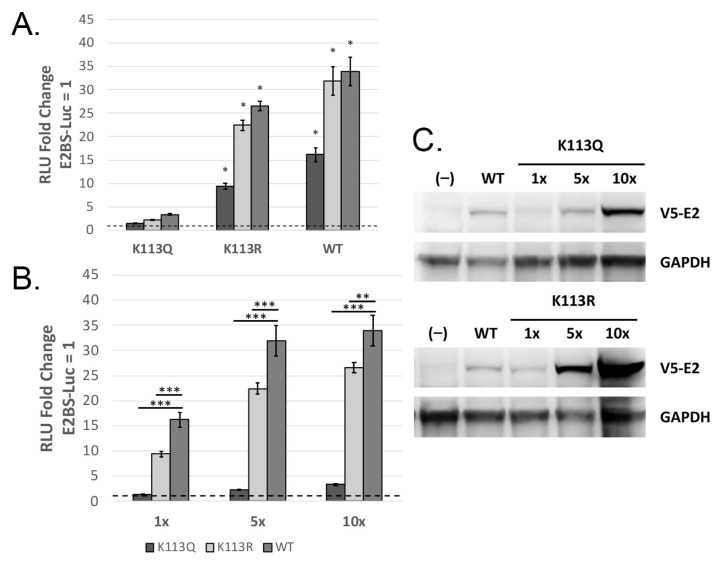
K113Q is transcriptionally defective. (**A**) K113Q is unable to activate transcription to a significant level, even when over-expressed ten-fold. K113R and WT E2 can activate transcription in a dose-dependent manner when over-expressed. * *p* < 0.05, Dunnett’s test. E2 vector amounts are titrated as 1× (50 ng), 5× (250 ng), and 10× (500 ng). (**B**) The data in panel A are presented and analyzed for comparison against WT E2. Data are grouped by expression level to demonstrate that K113R transcription is significantly diminished compared to WT E2. *n* = 3, * *p* < 0.05, ** *p* < 0.01, *** *p* < 0.001, paired *t*-test. (**C**) Immunoblot of whole cell lysate. Both K113 mutants display decreased protein levels compared to WT E2 at the 1× expression level, but protein levels increase when the mutants are over-expressed. Mouse anti-V5 (1:3000), mouse anti-GAPDH (1:8000).

**Figure 4 pathogens-14-00084-f004:**
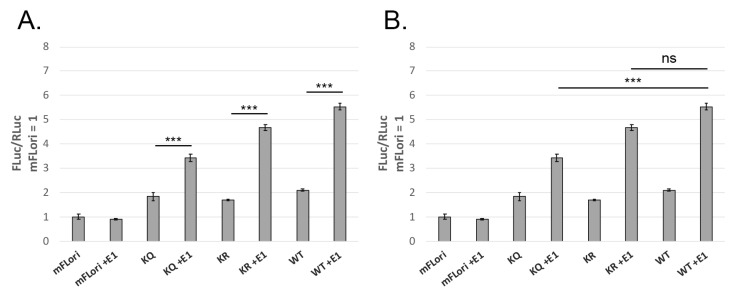
K113Q is deficient for transient replication. C33-A cells were transfected with mFLori, pRLuc, WT or K113 mutant pCI-V5-E2, and WT or TTL pCI-myc-E1. Luminescence was measured 72 h post-transfection. (**A**) Both mutants can transiently replicate compared to samples without E1. (**B**) Comparisons were made between WT and K113 mutant samples containing E1. K113Q showed a significant defect compared to WT E2. K113R consistently replicated to lower levels than WT E2, but this difference was not significant. *n* = 3, *** *p* < 0.001, paired *t*-test.

**Figure 5 pathogens-14-00084-f005:**
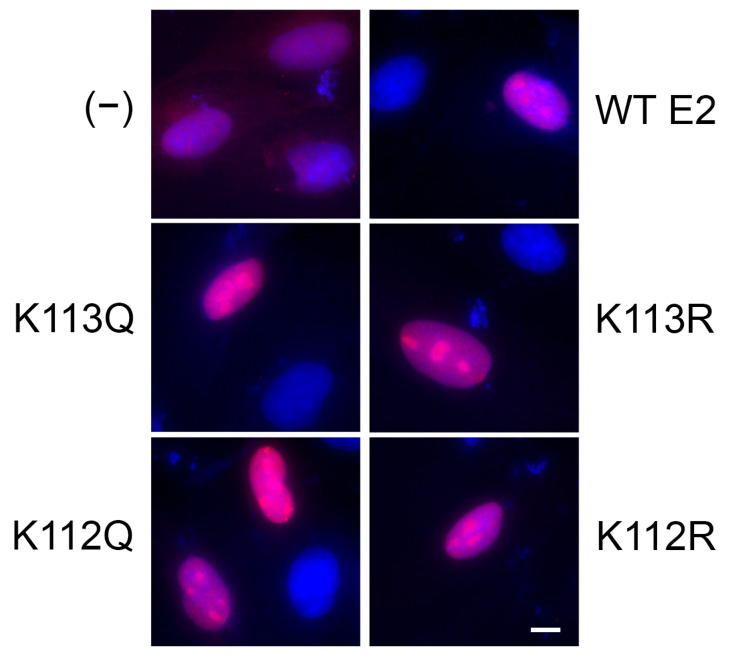
Mutation of the di-lysine motif does not interfere with nuclear localization. CV-1 cells were transfected with 3 μg of WT or mutant pCI-V5-E2. Cells were fixed 48 h post-transfection and stained for V5-E2 (Rabbit anti-V5, 1:1200; AlexaFluor 594 1:3000) and mounted with DAPI. Slides were analyzed for localization of E2 (red) within the nucleus (DAPI). Cells transfected with an mCherry expression vector were used as a control demonstrating diffuse localization. All di-lysine mutants were localized to the nucleus. Scale bar represents 10 μM.

**Figure 6 pathogens-14-00084-f006:**
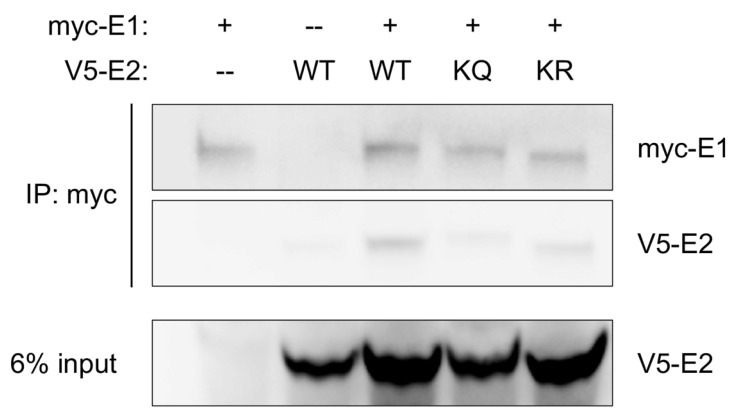
K113Q shows diminished binding to E1. HEK293TT cells were transfected with expression vectors for myc-E1 and WT, K113Q, or K113R V5-E2. Lysates were co-immunoprecipitated with anti-myc magnetic beads and protein complexes were separated by SDS-PAGE. WT (lane 3) and K113R (lane 5) immunoprecipitated with E1, while K113Q (lane 4) shows decreased binding. Immunoblotted with rabbit anti-Myc tag (1:3000) and rabbit anti-V5 (1:3000).

**Figure 7 pathogens-14-00084-f007:**
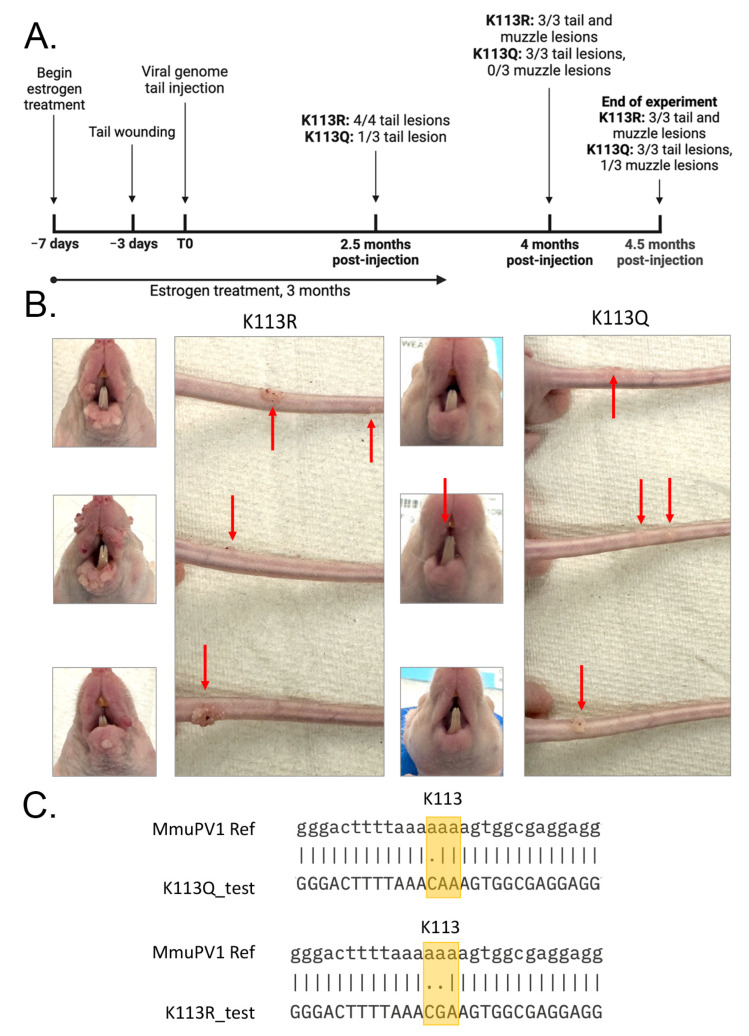
Both K113Q and K113R can induce wart formation in vivo. (**A**) Timeline of disease progression. Athymic hairless mice were injected intradermally with approximately 20 μg of circular mutant genomes and monitored weekly for the development of cutaneous lesions. K113Q lesion development lags behind K113R throughout the experiment. (**B**) Muzzle and tail lesions at the time of sacrifice, 4.5 months post-injection. K113R mice demonstrate florid lesion growth on the muzzle indicative of secondary infection, while only one K113Q mouse has developed a lesion on the muzzle. (**C**) Sequence alignment between DNA isolated from collected lesions and the MmuPV1 reference genome. K113 is highlighted in yellow, confirming that the lesions do contain the indicated E2 mutation.

**Figure 8 pathogens-14-00084-f008:**
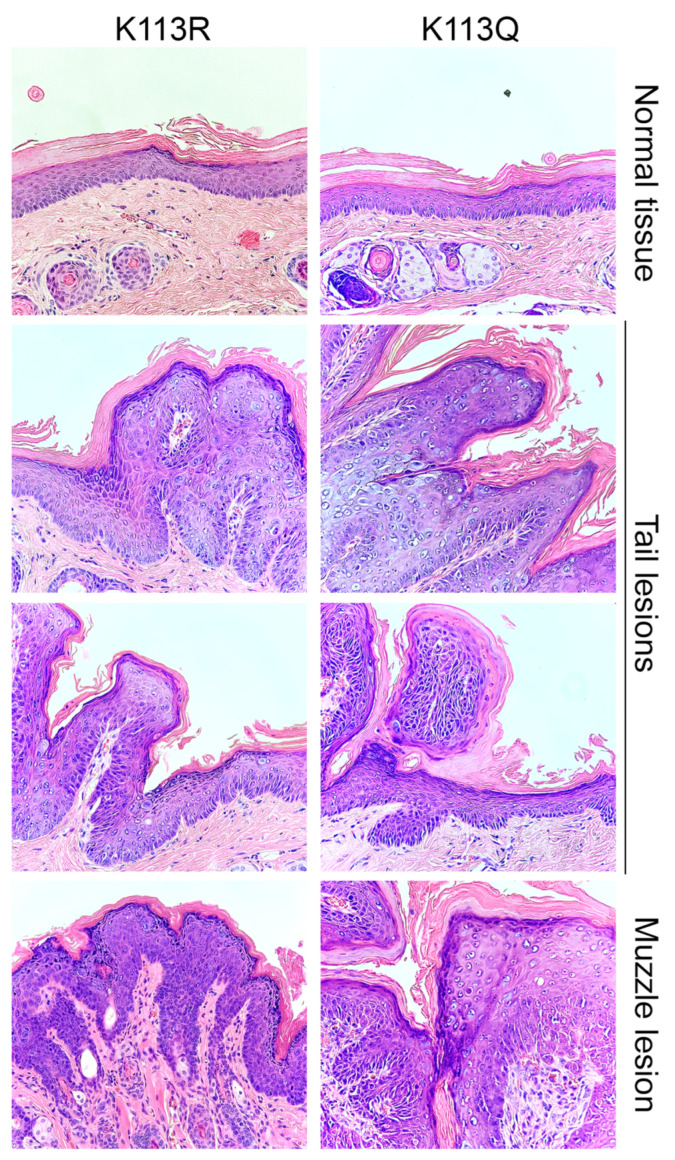
K113Q and K113R induce dysplasia in mice. Lesions were collected from the tail and muzzle and formalin fixed for H&E staining. The top panels are tissue sections without visible lesions and display normal histologic features. The lesions collected from both anatomic sites demonstrate histologic changes characteristic of PV-induced disease, including hyperkeratosis, parakeratosis, and koilocytic change (Images were taken at 20× magnification).

## Data Availability

The data presented in this study are available upon request from the corresponding author.

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
