# Peer review of "A Conserved Di-Lysine Motif in the E2 Transactivation Domain Regulates MmuPV1 Replication and Disease Progression"

_pathogens, 2025, doi:10.3390/pathogens14010084_

Round 1

Reviewer 1 Report

Comments and Suggestions for Authors

Gonzalez et al. describe the mutational analysis of two conserved consecutive lysines in the transactivation domain of the Mus musculus papillomavirus 1 E2 protein in vitro and in vivo. Using reporter assays they find that K112Q or A completely lose whereas K113Q or A largely retain activity. Both E2 K113Q and A are able to induce tail lesions in nude mice. This paper adds important information to E2´s role in vivo. I feel that the manuscript should be improved as some findings need more confirmation and more experimental details should be added.

1)      Fig. 2C: the western blots should be improved as wt E2 is barely visible. Wt E2 should also be titrated.

2)      1x, 5x, 10x should be replaced with actual amounts of plasmids

3)      Fig. 2E and 5: Immunofluorescence analysis need to be shown in the same cells (C33-A) as the reporter assays not 293TT or CV-1

4)      Why were no murine cells and MMuPV1 promoter constructs used?

5)      Fig. 7: it is difficult to draw any conclusions from the experiment without a wt control.

6)      The analysis of viral parameters (transcripts, copy number, no. of cells with amplified viral genomes, E4 and/or L1 protein expression) should be included

7)      No minus estrogen control is shown for the in vivo experiments. Does estrogen influence E2´s or E1´s activities?

8)      The replication constructs should be described in detail. l.103/104 “… were previously described (manuscript submitted)..” does not help the reader

9)      l. 125/126 AlexaFluor 488 and 594 are dyes and not secondary antibodies

Author Response

Comment 1:  Fig. 2C: the western blots should be improved as wt E2 is barely visible. Wt E2 should also be titrated.

Response 1: This figure has been adjusted to more easily visualize WT E2. Because both mutants are expressed less strongly than WT E2, titration of E2 would make it difficult to observe the dose-dependent increase in expression of the mutant proteins. The point of this figure is to show that increasing amounts of the mutant vectors leads to increased expression, so the most relevant data are prioritized.

Comment 2: 1x, 5x, 10x should be replaced with actual amounts of plasmids

Response 2: We did this to make the figures less crowded and easier to read. We have clarified plasmid amounts in the figure legends to address this comment.

Comment 3: Fig. 2E and 5: Immunofluorescence analysis need to be shown in the same cells (C33-A) as the reporter assays not 293TT or CV-1

Response 3: C33-A cells are not an ideal cell line for examining nuclear localization due to the small size and tendency to grow as dense colonies. We have verified in unrelated experiments that WT MmuPV1 E2 is nuclear in C33-A cells, and relevant papers by Quinlan et al. (2013) and Thomas and Androphy (2018) have not suggested that localization varies by cell type.

Comment 4: Why were no murine cells and MMuPV1 promoter constructs used?

Response 4: The transcription and replication assays utilized in this paper are well supported in the literature and are traditionally performed in C33-A cells. Experiments using primary murine keratinocytes were attempted but the signal was too low to be interpreted due to the poor transfectability of these cells.

Comment 5: Fig. 7: it is difficult to draw any conclusions from the experiment without a wt control.

Response 5: As stated in the text and reported by others, WT MmuPV1 virus can spread readily and rapidly in laboratory animal facilities, so the exclusion of a WT control in this set of experiments was an intentional infection control measure. The main takeaway from this experiment is that the replication defect displayed by K113Q in vitro did not prevent wart formation in vivo, indicating the completion of the productive stage of the viral life cycle.

Comment 6: The analysis of viral parameters (transcripts, copy number, no. of cells with amplified viral genomes, E4 and/or L1 protein expression) should be included

Response 6: It is unclear which figure this comment is referring to. If it is meant for the in vivo experiments, these analyses were not performed as they were not our intended endpoint for this study.

Comment 7: No minus estrogen control is shown for the in vivo experiments. Does estrogen influence E2´s or E1´s activities?

Comment 7: We are not aware of any direct effects that estrogen has on E1 or E2. It has been previously reported by others that estrogen increases MmuPV1 copy number (PMID: 36917668) and was used as a cofactor for wart development.

Comment 8: The replication constructs should be described in detail. l.103/104 “… were previously described (manuscript submitted)..” does not help the reader

Response 8: The referenced manuscript has now been published and details the constructs. Citation added.

Comment 9: l. 125/126 AlexaFluor 488 and 594 are dyes and not secondary antibodies

Response 9: This has been corrected.

Reviewer 2 Report

Comments and Suggestions for Authors

In this manuscript, Gonzales et al. demonstrate novel findings which suggest that the first lysine (K) residue within the motif, K112, is indispensable for E2-mediated transcription and transient replication in vitro.  Importantly, the findings also imply that acetylation of K113 may be involved in repressing MmuPV1 E2 activity.

This manuscript provides novel information about E2 protein activity.  I have only minor comments, which should be addressed in order to increase the quality of the manuscript.

Figure 2 E.  The IF images are blurry and should be replaced with better representative images.  Also the way in which they were obtained should be elaborated.  Microscope type etc.  If a confocal microscope was used scale bars should be also added.

Figure 5. Similar comment as for Figure 2.  The images, in particular (-), are blurry and should be replaced with better quality images.  Information about the way how these images were obtained should be included.  

Figure 8.  Scale bars are missing and more information on how these images were obtained should be included.

Author Response

Comment 1: Figure 2 E.  The IF images are blurry and should be replaced with better representative images.  Also the way in which they were obtained should be elaborated.  Microscope type etc.  If a confocal microscope was used scale bars should be also added.

Response 1: The materials and methods section has been updated to include more information on how the IF images were obtained. These are not confocal images. The images were sharpened to address blurriness.

Comment 2: Figure 5. Similar comment as for Figure 2.  The images, in particular (-), are blurry and should be replaced with better quality images.  Information about the way how these images were obtained should be included.  

Response 2: The image in (-) is demonstrating diffuse localization of the fluorescent protein and as such is expected to look this way. The materials and methods section has been updated.

Comment 3: Figure 8.  Scale bars are missing and more information on how these images were obtained should be included.

Response 3: The materials and methods section has been updated to clarify how these images were obtained.